# Design and Implementation of Edge-Fog-Cloud System through HD Map Generation from LiDAR Data of Autonomous Vehicles

**Junwon Lee, Kieun Lee, Aelee Yoo**  **and Changjoo Moon \***

Department of Smart Vehicle Engineering, Konkuk University, Seoul 05029, Korea; jileejun@konkuk.ac.kr (J.L.); lke1119@konkuk.ac.kr (K.L.); aelee1010@konkuk.ac.kr (A.Y.)

**\*** Correspondence: cjmoon@konkuk.ac.kr

**Abstract:** Self-driving cars, autonomous vehicles (AVs), and connected cars combine the Internet of Things (IoT) and automobile technologies, thus contributing to the development of society. However, processing the big data generated by AVs is a challenge due to overloading issues. Additionally, near real-time/real-time IoT services play a significant role in vehicle safety. Therefore, the architecture of an IoT system that collects and processes data, and provides services for vehicle driving, is an important consideration. In this study, we propose a fog computing server model that generates a high-definition (HD) map using light detection and ranging (LiDAR) data generated from an AV. The driving vehicle edge node transmits the LiDAR point cloud information to the fog server through a wireless network. The fog server generates an HD map by applying the Normal Distribution Transform-Simultaneous Localization and Mapping(NDT-SLAM) algorithm to the point clouds transmitted from the multiple edge nodes. Subsequently, the coordinate information of the HD map generated in the sensor frame is converted to the coordinate information of the global frame and transmitted to the cloud server. Then, the cloud server creates an HD map by integrating the collected point clouds using coordinate information.

**Keywords:** fog computing; big data platform; HD map; IoT; Hadoop ecosystem; NDT mapping

---

## 1. Introduction

The Internet of Things (IoT) is a network of objects with embedded sensors for exchanging data over the Internet [1]. IoT is utilized in various areas, including industrial and infrastructure systems, in addition to personal environments. IoT technology is also leveraged in self-driving vehicles (i.e., autonomous vehicles, AVs). Self-driving vehicles collect driving environment data using various systems, such as cameras, light detection and ranging (LiDAR), radars, and the global positioning system (GPS), and can operate as an IoT edge computer. According to a survey by the Hyundai Mobis Research Institute, an AV produces data at a rate of approximately 4 TB/h. In addition, within the next 5–10 years, 90% of US cars are expected to be replaced by self-driving cars with 1–3 levels of autonomous driving functionality, and the amount of data generated is expected to increase further [2]. Therefore, when multiple AVs operate simultaneously, the corresponding data will fall within the Big-Squared-Data space, which is outside the scope of big data. Therefore, a new efficient data processing platform will be required [3].

Vehicles connected through a network operate in a similar manner to that of IoT devices and require a central management system to analyze and process the data generated from the multiple terminals. The existing architectures that act as central servers for processing big data are illustrated in Figure 1. As depicted, two main architectures exist. Because every platform has its own set of strengths

and weaknesses, it is necessary to select the most appropriate architecture according to the goals of the system to be built.

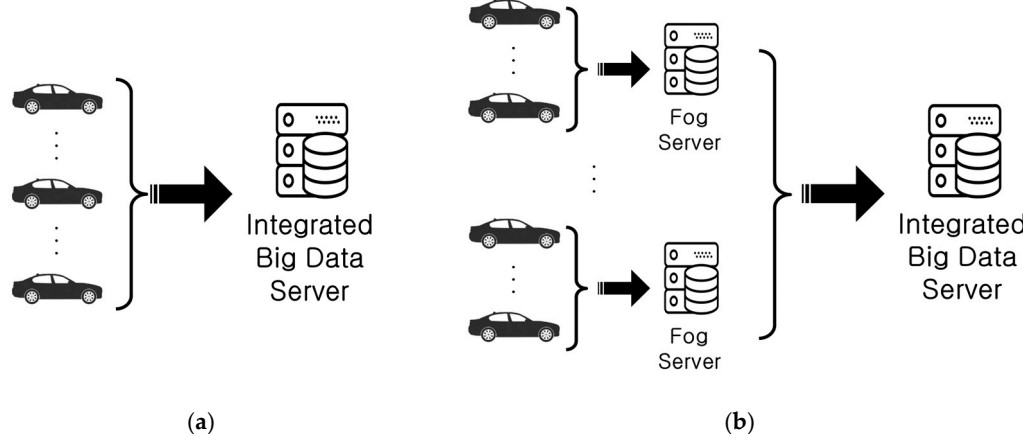

**Figure 1.** Cloud server vs. fog server. (**a**) Concept of cloud server; (**b**) Concept of fog server.

The cloud server architecture depicted in Figure 1a is suitable for a system that processes a number of light operations quickly because the cloud server manages all of the data; examples include banking and text messaging systems. In addition, this cloud-based architecture has an advantage pertaining to data loss because it is easy to identify the terminal corresponding to the error source. However, because all of the data are concentrated on the central server, there is a delay due to the limitation of processing resources [4].

To compensate for these problems, Bar-Magen Numhauser proposed fog computing in 2012 [5]. Fog computing is a network structure for IoT equipment. As depicted in Figure 1b, a small cloud server called a fog server is installed between the central server and the vehicle terminal. This structure has the advantage of parallelizing several complex processes. Because each fog server processes the data generated by the terminal and sends the processing result to the cloud server, the delay caused by one process monopolizing the central server is reduced.

In this study, we propose an integrated Hadoop ecosystem–fog server system for big data analysis that processes LiDAR big data generated from AVs to produce high-precision maps required for autonomous driving. It connects the data source and the fog server through the robot operating system (ROS) to collect LiDAR information generated from multiple AVs that are the end nodes of the system. The fog server processes the LiDAR point cloud data according to the NDT algorithm. In the fog server, each point cloud is indexed into a three-dimensional (3D) space and a voxel space; subsequently, the point cloud inside each voxel is approximated through a 3D Gaussian distribution to obtain the point cloud of a fixed object. The cloud server combines the high-definition (HD) map information transmitted from multiple fog servers into one and transmits the final generated HD map to the big data server. The Kafka message system provides a real-time data streaming environment connecting the HD map-generation system built in the ROS environment to the big data system built in the Hadoop environment. Map information is uploaded to the ROS environment through the Kafka server, and downloaded from the Hadoop environment, and large-scale HD maps are distributed and stored in the Hadoop distributed file system (HDFS).

The remainder of this paper is organized as follows. A description of previous studies on big data systems that process LiDAR data is provided in Section 2, followed by the overall design of the fog computing system based on the architecture of the edge-fog-cloud system and selection of the software platform used. Section 3 outlines the establishment of the edge-fog-cloud system and the setting of the development environment based on the data collected at Konkuk University. Section 4 presents an HD map based on the designed and implemented system, and verifies whether the big data system is operational. The conclusions of this study are presented in Section 5.

## 2. Design of Fog Computing System

In this section, we discuss the previous studies that efficiently handled point cloud data using a big data platform. In addition, we design the edge-fog-cloud system for large-scale HD map generation and determine the software platform for system configuration.

### 2.1. Related Research

Research on the storage and utilization of the LiDAR-generated point cloud data in a big data platform is ongoing.

Vo et al. developed an encoding technique to rapidly upload point cloud data to a big data platform [6]. Deibe et al. proposed an architecture that reduces the processing time by storing the data in an aviation LiDAR point cloud big data platform database [7]. Malik et al. utilized measured terrain models to quickly process point clouds [8]. The commonality in these studies suggests that processing point cloud data using a big data platform is effective. The limitation is the fact that the big data platform processes a single static point cloud data, rather than processing data from multiple data sources in real time.

### 2.2. Edge-Fog-Cloud System Architecture

The fog computing-based big data processing system physically stores data in a big data server through three stages of computing. The main components of this architecture are an edge node, a fog computing server, and a cloud computing server. The system architecture is illustrated in Figure 2.

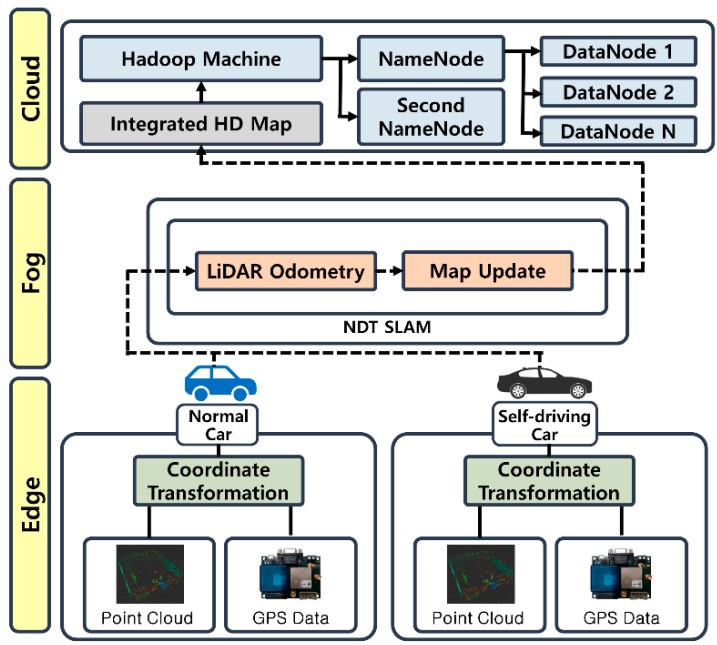

**Figure 2.** Edge-fog-cloud architecture.

### 2.2.1. Edge Node

The edge node is a computer that collects the information obtained from the sensor attached to each vehicle. It is located at the lowest level of the edge-fog-cloud system. There are three main tasks handled by the edge node computer: collecting LiDAR data through a LiDAR driver, collecting location information through a GPS device, and communicating with the fog computing server using the ROS. The edge node computer does not require any high-level computing power because it transmits the LiDAR and GPS data relayed by the sensors to the fog computing server through the network. Each

AV acts as an edge node and facilitates data collection by transmitting the sensed data, rather than resources for calculation.

### 2.2.2. Fog Server

The fog server performs pre-processing of the LiDAR point cloud data generated in the edge nodes. The fog server receives the point cloud and location information data from the LiDAR- and GPS-equipped edge nodes. It carries out the following two major operations. First, the point cloud data are analyzed to distinguish between the fixed obstacles and moving objects; through filtration, only the fixed obstacle point cloud information necessary for the HD map generation is retained. Subsequently, as the point cloud is created based on the LiDAR data at each edge node, a coordinate system transformation operation is performed to represent all of the point clouds in one frame. Unlike the edge nodes that collect information, the fog server requires a considerable amount of computational resources because it processes the data generated by multiple edge nodes.

### 2.2.3. Cloud Server

The cloud computing server facilitates transformation of the point cloud data processed by the fog computing server into one coordinate plane, and finally generates the HD map information. Saving and managing a large number of HD maps in an ROS environment to create map data is inefficient in terms of data loss and utilization. For this reason, a big data processing platform, HDFS, is deployed inside the cloud server. The HDFS architecture consists of the DataNode, where the distributed data are stored; the NameNode, which manages the location of the data distributed through metadata; the secondary NameNode, which replicates the data of the NameNode to prepare for failure; and the Hadoop machine, which is responsible for the overall management of the Hadoop big data platform.

### *2.3. Software Platform Structure*

To build a fog computing-based big data processing system, we used the Ubuntu Linux-based ROS that performs LiDAR point cloud data processing and the Hadoop ecosystem to store and utilize the large-scale HD maps generated in the CentOS Linux-based HDFS. In addition, the cloud computing server uses Apache Kafka to communicate between ROS and CentOS Linux and the other two Linux systems [9].

### 2.3.1. ROS Message

The edge nodes, fog computing servers, and cloud computing servers use ROS to process the LiDAR point cloud; the communication between each machine is performed through the ROS [10]. Message topic management is performed as depicted in Figure 3.

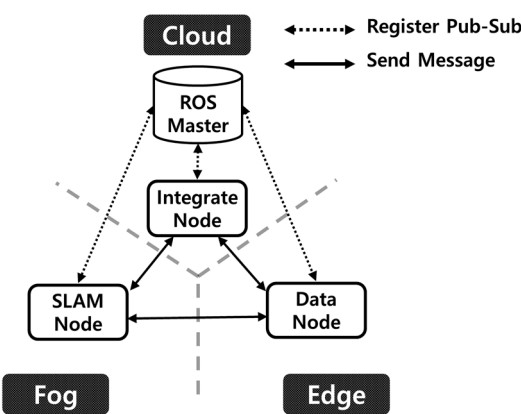

**Figure 3.** Concept of robot operating system (ROS) message system communication.

It is possible to check the IP address and port number of the running node, in addition to the contents of the topic being generated through the ROS message system. This feature allows one system to operate organically in multiple machines. The ROS master is required to execute ROSCORE and manage the messages to run the ROS message system. The ROS master in a cloud computing server ensures that all of the terminal machines can access a fixed location and share the information on sending and receiving messages [11].

### 2.3.2. Kafka Message System

A message system is required to handle the messages generated in ROS in the Hadoop environment built on CentOS. This study utilizes Kafka, a real-time asynchronous streaming solution. The structure of the Kafka message system comprising a message server is depicted in Figure 4.

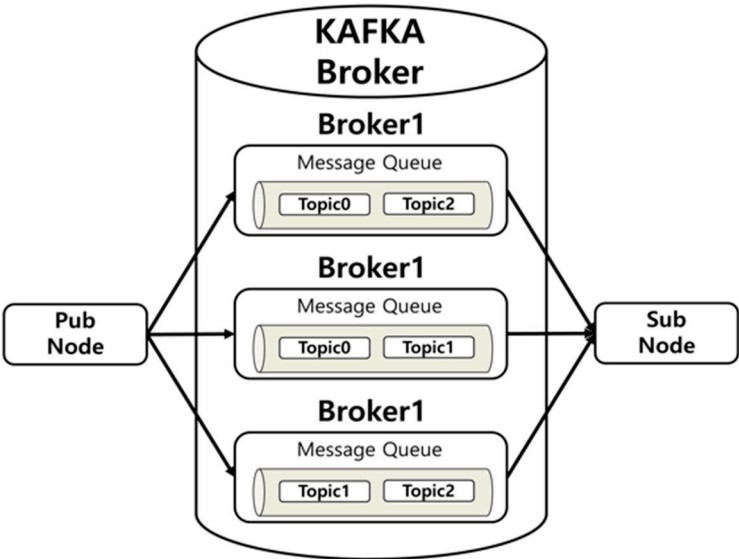

**Figure 4.** Kafka message system communication.

The Kafka broker server is located between the pub node that produces messages and the sub node that subscribes to messages to relay messages. The Kafka broker builds an internal message queue to process incoming First In First Out (FIFO) messages. The configuration of a broker server using multiple machines rather than a single broker is depicted in Figure 4. Because multiple brokers can replicate and store the same message, even if one broker stops operating due to a failure during server operation, the system still remains operational without any loss of messages.

The most advantageous Kafka system is a structure consisting of multiple instances of producer–broker–consumers, as illustrated in Figure 5.

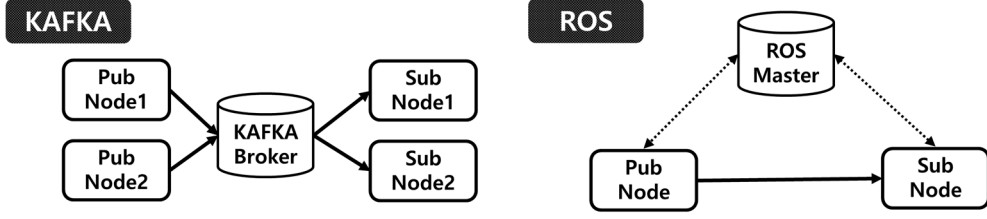

**Figure 5.** Kafka message system vs. ROS message system.

Unlike the ROS message system described above, a 1:N or N:N connection is formed in the Kafka message system by linking the producer and subscriber through a broker, rather than a 1:1 direct

connection between the data-generating node and the subscriber node. Furthermore, this connection method also reduces the issuance cycle of messages; when the system is expanded, it has the advantage of communication via a simple connection with the Kafka broker. In addition to the aforementioned advantages of the Kafka message system, the Kafka library facilitates communication regardless of the development environment.

### 2.3.3. Hadoop Ecosystem

HDFS is used to efficiently store the created large-sized point cloud files. The system configuration is depicted in Figure 6. It deploys two Hadoop ecosystem applications, Hadoop and Yarn [12,13].

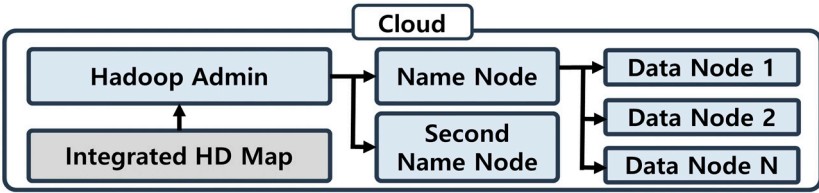

**Figure 6.** Hadoop cluster.

HDFS consists of a client, NameNode, and DataNode. Hadoop Admin is connected to the integrated HD map node in the Ubuntu environment, and it receives HD map information and creates commands to store the data in HDFS. The NameNode and secondary NameNode serve as storage to store the location of the data to be saved. If the location information of the data existing in the NameNode is distorted or lost, the location of the DataNode where the data are stored cannot be known. Therefore, the same information stored in the NameNode is also stored in the secondary NameNode and backed up [14]. The DataNode facilitates physical storage of data, along with a large HD map that is divided prior to storage. Because it is a distributed file system, if the storage space needs to be expanded, horizontal expansion is performed by adding a machine that acts as a physical data node.

## 3. Real-Time HD Map Generation Based on Multiple Edges

Because the implemented fog computing architecture is optimized for fast computation by distributing large amounts of data, it is suitable for processing both image and LiDAR data, in addition to the other types of data obtained from the AVs. The primary objective of this study is to build a large-scale HD map using a fog computing environment; therefore, LiDAR data are the primary focus.

### 3.1. Dataset

LiDAR data were obtained as point cloud data using the Velodyne VLP-16 sensor of Velodyne Lidar [15]. The location information was collected using a Ublox C94-M8P device of U-Blox to represent all of the coordinate points of the point cloud in terms of a single global coordinate system. To generate a real-time HD map, the road within Konkuk University was divided into nine sections, and the data corresponding to each driving route were collected, as shown in Figure 7. Each set of driving data comprised the driving data for 5 min over a distance of approximately 1–2 km, and a total of 13 GB of data was obtained.

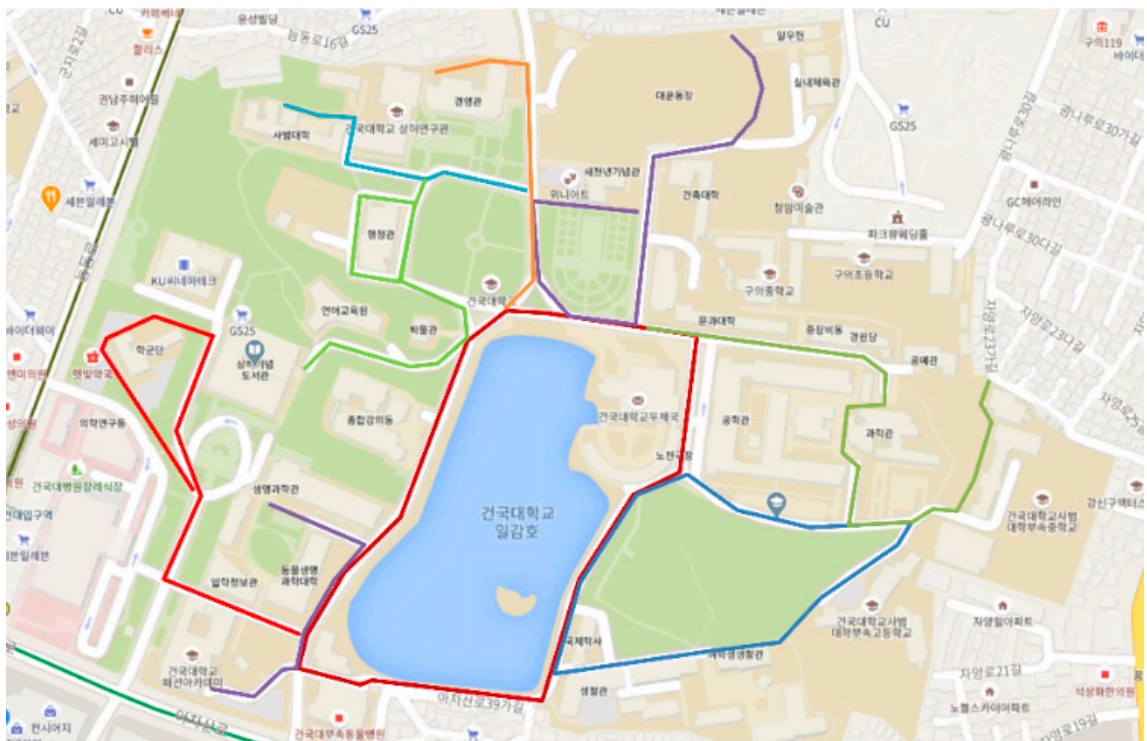

**Figure 7.** Driving route plan.

### 3.2. NDT-SLAM

The ICP algorithm is typically used to define the point cloud set [16]. However, due to the nature of the algorithm that repeats the operation of measuring the correspondence distance between the point cloud and previous data each time and moving the point cloud through the transformation matrix, there are disadvantages such as lengthening the operation and causing an error according to the initial point. The NDT-SLAM technique overcomes this problem by matching the maps using statistical characteristics [17]. The NDT algorithm facilitates fast computation speed by processing the voxel spatial indexing in 3D space and approximating the point cloud inside each voxel to a 3D Gaussian distribution. The NDT-SLAM algorithm was implemented using the NDT-Mapping algorithm of Autoware, an open source software suite for autonomous driving simulation managed through Tier IV in Japan [18].

### 3.3. Deployment of Edge-Fog-Cloud System

When deploying a large-scale precision map-generation system using collected data, the following are necessary: a system design for sending and receiving data processed in a distributed computing environment, a data transmission method between operating systems (OSs) rather than a single OS through Kafka, and a Hadoop ecosystem configuration for a distributed environment.

#### 3.3.1. Deployment of Edge-Fog-Cloud System

The hardware system configured in this study is shown in Figure 8. The nine edge nodes in the figure integrate the LiDAR and GPS data collected from the AV and transmit it to the fog server.

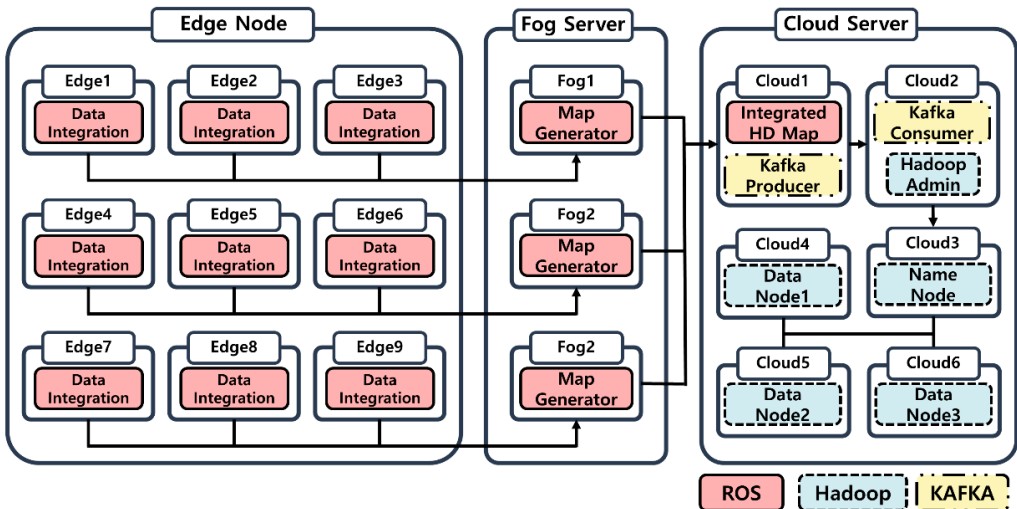

**Figure 8.** Edge-fog-cloud system.

Each fog server performs map generation on the collected data and sends the generated HD map to the cloud server. The edge nodes then collect the vehicle data send them to the Map Generator, which creates an integrated HD map. The point cloud is built in the ROS environment, and then Kafka Producer and Consumer executes data communication between ROS and Hadoop. Hadoop Admin is placed on the same machine as that of the Kafka consumer in the cloud server, and the environment is constructed so that the received data can be distributed and stored after stream processing. The machine where Hadoop Admin is installed communicates with the machine that acts as a NameNode and establishes a system to distribute and store HD maps in the data nodes installed in Cloud 4–6.

3.3.2. Configuration of Development Environment

To deploy the edge-fog-cloud system in a distributed environment, the OS and software platform are configured as shown in Figure 9.

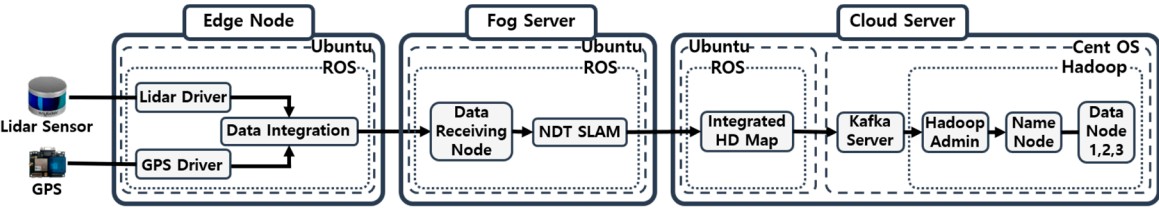

**Figure 9.** Schematic diagram of edge-fog-cloud system functionality.

Ubuntu 18.04 LTS and Cent OS 7 were used as the OSs. The reason for using multiple OSs is because the driven system determines the required characteristics of the OS. Ubuntu is a Debian-series Linux system that focuses on regular updates rather than system stability, and has the advantage that it quickly reflects new technologies. However, it has a shortcoming in terms of stability because the period for version support is short. In contrast, CentOS is Red Hat-series open software, and is used as a server OS in many organizations due to its stability. Based on the role that cloud technology plays in this study, CentOS is particularly suitable because it focuses on the integrity of data, despite having the limitations of storage and processing server functionalities. In the case of edge and fog servers, ROS, the middleware used for sensor data processing, is deployed in the Ubuntu environment.

### 3.3.3. Configuration of Edge Node

The edge node is accessed in the Ubuntu-based ROS environment, and three ROS nodes are running. The ROS-Velodyne LiDAR driver is used to convert the LiDAR data into the rostopic format [19]. The GPS data are also converted into the rostopic format using a GPS driver package published through GPS ros.org [20]. The two topics created synchronize two topics with different occurrence times into one topic through the data integration node and transmit them to the fog server. Through this pre-processing, multiple edge nodes can process data with minimal computing resources.

### 3.3.4. Configuration of Fog Server

The data generated by the multiple edge nodes are transmitted to the fog server and subjected to pre-processing for map generation. In the case of duplicate topic names in the ROS system, the system stops. Therefore, the unique number of each edge node is assigned to each topic to prevent duplication of the name, and the input is inserted into the NDT-SLAM node. The NDT-SLAM algorithm is implemented using the NDT-Mapping algorithm of Autoware.

### 3.3.5. Configuration of Cloud Server

The results of the NDT-SLAM precision maps generated by the multiple fog servers are transmitted to the cloud server and integrated into a single map. As an environment for ROS operation, which is an environment that generates maps, the generated map data must be stored in a big data platform. Two different systems configure a Kafka server for real-time data streaming, as shown in Figure 9. The integrated HD map node of the ROS environment in Figure 9 plays the role of a producer that creates a Kafka message while integrating the map and transmitting the generated data to the defined Kafka server. The data are accumulated in the server exchange data in a push-pull method, and the data can be converted into data that can be used in the big data platform through the Hadoop Admin node based on Hadoop in the CentOS environment. HD map data delivered through the Hadoop Admin machine are stored in HDFS. Because the Hadoop system is a distributed processing system composed of a number of machines rather than a single machine, the location of the DataNode storage and data distribution is processed through the NameNode.

### 3.4. Autonomous Vehicle Security System Based on Edge-Fog-Cloud

When the vehicle's driving data are collected in the fog and cloud servers, a privacy issue that threatens personal information arises. To prevent this problem, a security system must be built inside the system. The ROS environment that communicates data in the edge-fog-cloud system does not provide its own security techniques other than being careful not to disclose the information of the ROS master, which manages the information of the nodes sending and receiving data [21]. For this reason, encryption and decryption techniques according to the Advanced Encryption Standard (AES) are applied to solve the privacy security problem of the system to be built. In addition, the authentication method is applied when the edge node accesses the upper system by issuing an Auth-Token with OAuth2 so that unauthorized users cannot access it through the authentication system [22,23].

As shown in Figure 10, the edge node receives an access token from the fog server based on the access information of each vehicle. Vehicles that are permitted access through the token allocated to the edge node transmit the vehicle data information together with the token information to the server. Among the various data generated by a vehicle, GPS data are the most vulnerable to privacy issues because they contain user location information. Vision data do not directly affect the privacy of users, but they may lead to the violation of the privacy of others, such as pedestrians around the driver and information of other vehicles. The remaining sensor data, such as LiDAR, RADAR, and CAN data, are driving status data and do not significantly affect privacy. Thus, this information is not classified as sensitive information. When the original data are transmitted through the network, the data are exposed. After these data are transmitted following AES-based encryption and decryption in the edge

node, the original data can be recovered by decrypting the data in the fog server. The result is shown in Figure 11.

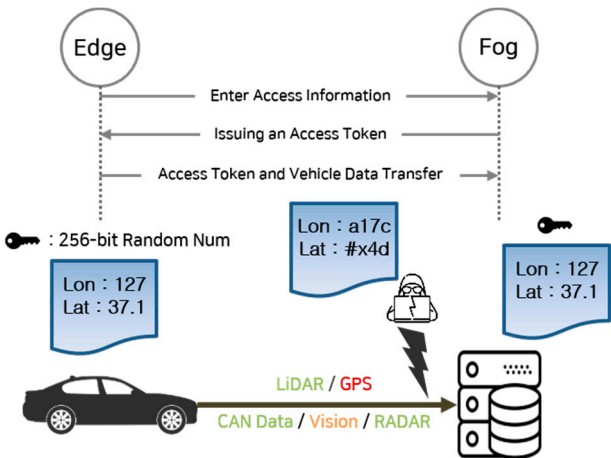

**Figure 10.** Sensor data security concept diagram of an autonomous vehicles (AV).

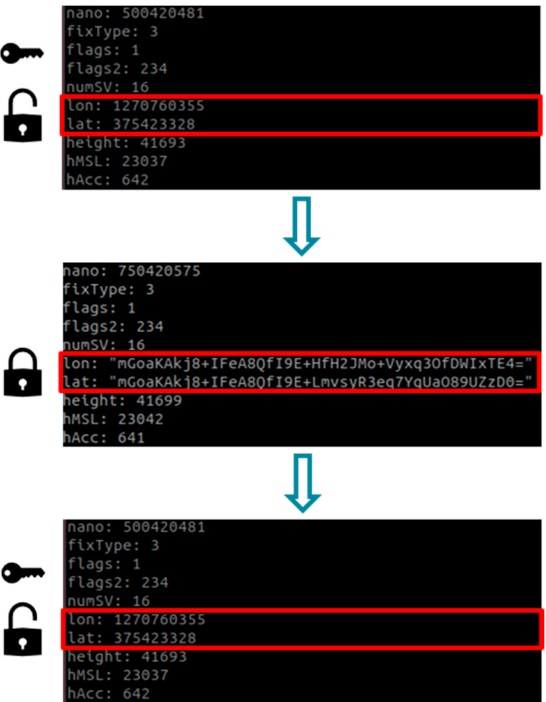

**Figure 11.** AV's sensor data encryption and decryption example.

## 4. Edge-Fog-Cloud-Based Map Creation and Storage

Real-time HD maps are created using the constructed system and data, and the generated maps are uploaded to the Kafka server. Data collected in the Kafka server are stored in the Hadoop cluster through Spark Streaming.

*Generation of HD Maps for Each Section*

Using the dataset collected for system verification, we generated an integrated HD map for nine sections. The nine sections are depicted in Figure 12.

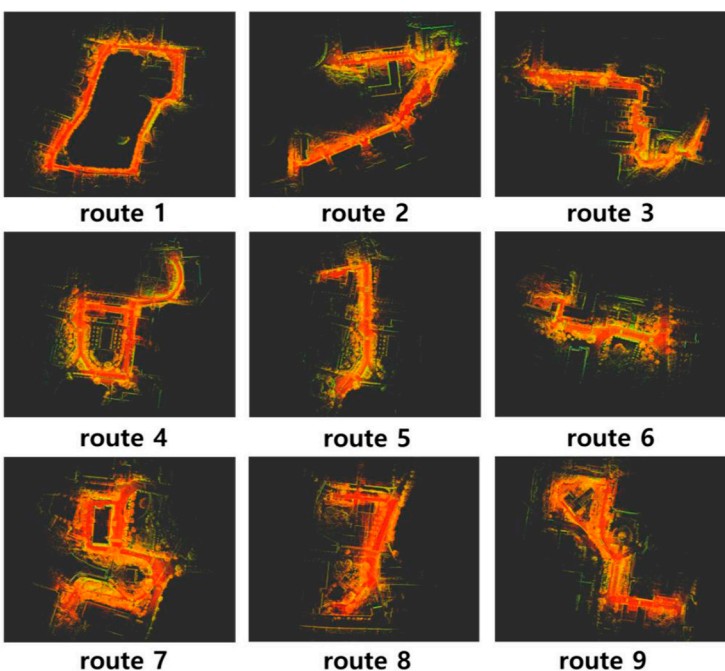

**Figure 12.** High-definition (HD) map generated for each route.

The HD maps corresponding to the data generated from edge nodes 1–3, 4–6 and 7–9 are generated by fog servers 1, 2 and 3 respectively, using the NDT-Mapping algorithm. The generated integrated HD map is illustrated in Figure 13.

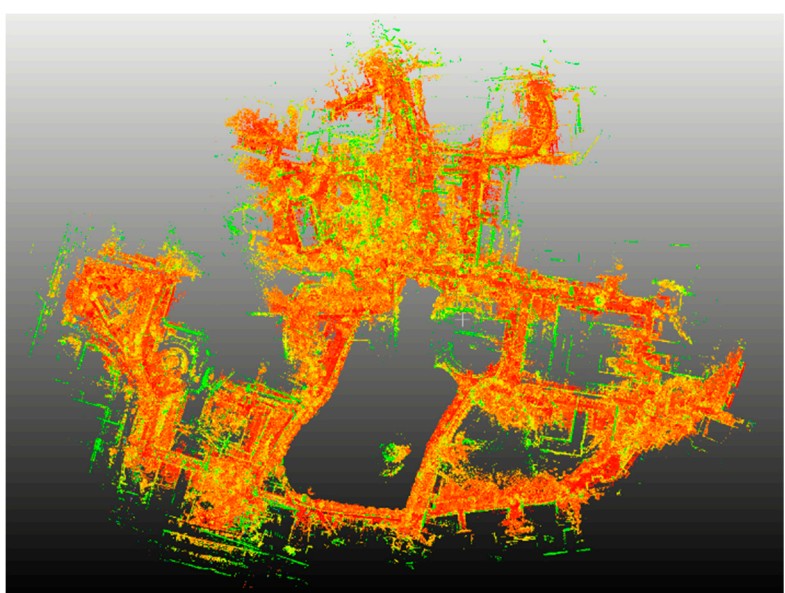

**Figure 13.** Integrated large-scale HD map.

The integrated HD map from the cloud ROS server is transmitted to the Kafka server and is pushed to the message queue structure in the order uploaded to the server. It distributes data in the resilient distributed dataset (RDD) format using Spark Streaming for storage in the Hadoop cluster.

The data collected in real time using Spark Dstream are stored in a variable in the RDD format, and the variable is distributed as a JSON file, as depicted in Figure 14. The distributed stored data can be reconstructed again and used to store and process large amounts of data.

```
-rw-r--r--   1 hadoop supergroup     885834 2020-09-16 13:02 /user/hadoop/saveOutput/part-00000-01174712-17fa-4edd-8208-0b6a7d476950-c000.json
-rw-r--r--   1 hadoop supergroup     898334 2020-09-16 13:02 /user/hadoop/saveOutput/part-00000-04865661-2ffb-457c-81c4-0ff28c916b93-c000.json
-rw-r--r--   1 hadoop supergroup     943434 2020-09-16 13:03 /user/hadoop/saveOutput/part-00000-055be2e5-f846-4a8c-a07f-ca4774476507-c000.json
-rw-r--r--   1 hadoop supergroup     927902 2020-09-16 13:03 /user/hadoop/saveOutput/part-00000-059e288c-85d5-438e-a46b-da9e2b7e1d9a-c000.json
-rw-r--r--   1 hadoop supergroup     898974 2020-09-16 13:02 /user/hadoop/saveOutput/part-00000-07b617bb-8241-4dab-85b2-2ed050edd31b-c000.json
```

**Figure 14.** Distributed stored HD map in DataNode.

## 5. Quantitative Evaluation

For quantitative evaluation of the established system, the processing time of the system according to the increase in processing data, the processing resources required for HD map generation, and the change in processing speed according to the increase in data size were evaluated. For the evaluation, Route 1 data were used; these are the driving data for a 1.2 km distance in the same nine edge nodes. The processing time according to the increase in the number of vehicles in the constructed system is shown in Figure 15.

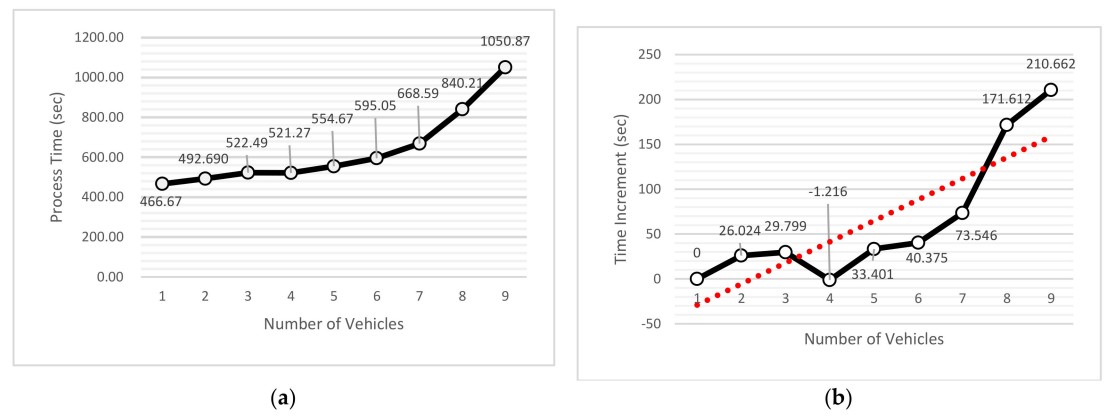

(**a**)　　　　　　　　　　　　　　(**b**)

**Figure 15.** Changes in processing time according to the number of vehicles. (**a**) Processing time according to the number of vehicles; (**b**) Change in processing time according to the number of vehicles.

As shown in Figure 15a, the processing time of the system increased from 466.6 s to 1050.9 s for one vehicle. Figure 15b is a graph of the increase in processing time when the number of vehicles increased from one to nine. It can be seen from the graph that the processing time also increases with the number of vehicles. The virtual memory usage according to the increase in the number of vehicles is shown in Figure 16a, and the processing speed according to the increase in the number of vehicles is shown in Figure 16b.

From the above graphs, it can be seen that the processing resources required by the system increase with the number of vehicles, and processing delay occurs. Do et al. [24] showed that it takes approximately 23 s to generate an HD map for a distance of 20 m using the traditional NDT-mapping algorithm, and 7.8 s when processing a point cloud for a distance of 20 m using the edge-fog-cloud system. The reason for this difference is that the map generation in this study is based on various variables, such as the movement speed and the size of the voxel when processing LiDAR data, although the difference in the method for generating the map and the map distance is the same. In creating HD maps, we believe that these performance indicators can be sufficiently used in a near real-time system.

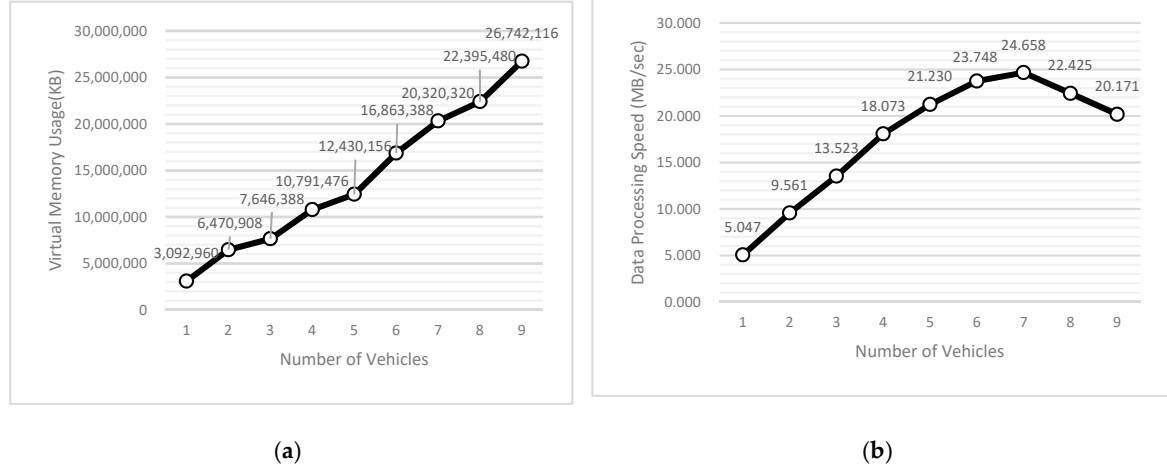

**Figure 16.** Changes in virtual memory usage and processing speed according to the number of vehicles. (**a**) Virtual memory usage according to the number of vehicles; (**b**) Processing speed according to the number of vehicles.

In addition, this study was designed with the goal of collecting data from multiple vehicles using an IoT platform, creating a HD map based on the collected data, and storing it in a big data platform. For this reason, the map-generation method using NDT-SLAM, the network communication method between platforms, and the performance of the server computer can be optimized and improved.

## 6. Conclusions

In this paper, we propose a system that processes point cloud data obtained from multiple AVs in real time using the edge-fog-cloud-based computing environment and stores the generated large-scale HD maps in a big data platform. To collect the point cloud and GPS information from the AVs, the system is configured using an Ubuntu-based ROS, and the sensor data storage function of the ROS is used to configure an edge node environment in which the vehicles on nine different paths generate data simultaneously. The fog server is also configured as a system in the ROS environment for processing LiDAR data, and one fog server collects data from multiple edges. Only the point cloud data of fixed objects are retained for HD map generation from the collected data using the NDT-Mapping algorithm. In addition, the generated point cloud data are transmitted to the cloud server by performing a coordinate system transformation to represent a plurality of point cloud data as a single reference point of the LiDAR point cloud data. A single large-scale HD map is created by merging the HD maps collected on the cloud server. A big data platform is required to manage the generated map; CentOS, which has the advantage of system stability, is used as the OS of the big data platform. The big data platform is implemented using the Hadoop ecosystem. For real-time communication between two different systems, the Kafka message system is used to process asynchronous messages in real time. The transmitted large amount of point cloud data is stored in the HDFS in the JSON format. A large-scale precision map is created with this message.

In the future, we intend to filter the data by querying the data in non-structured query language (NoSQL). NoSQL is a format for unstructured data, and is used for data analysis. In addition, the vehicle status information and image data information, similar to the LiDAR data, can also be processed using the big data platform.

**Author Contributions:** Conceptualization, J.L. and C.M.; methodology, J.L. and C.M.; software, J.L., K.L. and A.Y.; validation, J.L., K.L. and A.Y.; investigation, J.L. and K.L.; data curation, J.L. and K.L.; writing—original draft preparation, J.L.; writing—review and editing, J.L., K.L., A.Y. and C.M.; visualization, J.L.; supervision, C.M.; project administration, C.M.; All authors have read and agreed to the published version of the manuscript.

**Funding:** This research was funded by the Korea Institute for Advancement of Technology (KIAT) grant funded by the Korean Government (MOTIE) (N0002428, The Competency Development Program for Industry Specialist).

**Conflicts of Interest:** The authors declare no conflict of interest.

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
