# Peer review of "Design and Implementation of Edge-Fog-Cloud System through HD Map Generation from LiDAR Data of Autonomous Vehicles"

_electronics, doi:10.3390/electronics9122084_

Round 1

Reviewer 1 Report

The paper proposes a configuration that enables processing point cloud data from multiple data sources in real time. This is an extension of the existing solution that processes a single static point cloud data.

The new system was tested for nine cars simultaneously moving on different paths.  A single large-scale high definition (HD) map was created in real time by merging the HD maps collected based on LiDARs information from particular cars. Driving data for 5 min over a distance of approximately 1–2 km were used.

The structure of the system is clearly described and convincing.

However, I expect more detailed information on timing constraints of real time processing in your system.

Author Response

Thank you for advise me about the improvements to my paper.

The details of reflection on the improvements have been written in detail in the attached file.

Thank you.

Reviewer 2 Report

In this study, the authors propose a fog computing server model that generates a high-definition map using light detection and ranging (LiDAR) data generated from an autonomous vehicle and builds a test bed.

Self-driving cars or autonomous vehicles are of real interest in the current context of developments of IoT technologies and the increase in computing power of servers capable of processing very large amounts of data.

Considering that the application is intended for autonomous cars, my concern is related to the speed of real-time data processing, especially those from GPS, as well as their security. It should be specified in the paper what is the data processing speed (at different stages of the system) and whether it is sufficient for the autonomous car application.
Also, please specify in the paper what measures have been taken to ensure the security of the data processed by the proposed system. Ensuring data security is essential for autonomous car applications.

Author Response

(The authors gave the same response as above.)
